# Fabrication of Metallic Superhydrophobic Surfaces with Tunable Condensate Self-Removal Capability and Excellent Anti-Frosting Performance

**DOI:** 10.3390/nano12203655

**Published:** 2022-10-18

**Authors:** Jian-Guo He, Guan-Lei Zhao, Shou-Jun Dai, Ming Li, Gui-Sheng Zou, Jian-Jun Wang, Yang Liu, Jia-Qi Yu, Liang-Fei Xu, Jian-Qiu Li, Lian-Wen Fan, Min Huang

**Affiliations:** 1Aerospace Information Research Institute, Chinese Academy of Sciences, Beijing 100094, China; 2School of Optoelectronics, University of Chinese Academy of Sciences, Beijing 100049, China; 3Key Laboratory of Computational Optical Imaging Technology, Chinese Academy of Sciences, Beijing 100094, China; 4State Key Laboratory of Automotive Safety and Energy, School of Vehicle and Mobility, Tsinghua University, Beijing 100084, China; 5State Key Laboratory of Tribology, Key Laboratory for Advanced Manufacturing by Materials Processing Technology, Ministry of Education of PR China, Department of Mechanical Engineering, Tsinghua University, Beijing 100084, China; 6State Key Laboratory of Transient Optics and Photonics, Xi’an Institute of Optics and Precision Mechanics of CAS, Xi’an 710119, China; 7Institute of Chemistry, Chinese Academy of Sciences, Beijing 100190, China; 8Technology and Engineering Center for Space Utilization, Chinese Academy of Sciences, Beijing 100094, China

**Keywords:** ultrafast laser processing, surface nanostructuring, superhydrophobic, condensate self-removal, anti-frosting

## Abstract

Laser fabrication of metallic superhydrophobic surfaces (SHSs) for anti-frosting has recently attracted considerable attention. Effective anti-frosting SHSs require the efficient removal of condensed microdroplets through self-propelled droplet jumping, which is strongly influenced by the surface morphology. However, detailed analyses of the condensate self-removal capability of laser-structured surfaces are limited, and guidelines for laser processing parameter control for fabricating rationally structured SHSs for anti-frosting have not yet been established. Herein, a series of nanostructured copper-zinc alloy SHSs are facilely constructed through ultrafast laser processing. The surface morphology can be properly tuned by adjusting the laser processing parameters. The relationship between the surface morphologies and condensate self-removal capability is investigated, and a guideline for laser processing parameterization for fabricating optimal anti-frosting SHSs is established. After 120 min of the frosting test, the optimized surface exhibits less than 70% frost coverage because the remarkably enhanced condensate self-removal capability reduces the water accumulation amount and frost propagation speed (<1 μm/s). Additionally, the material adaptability of the proposed technique is validated by extending this methodology to other metals and metal alloys. This study provides valuable and instructive insights into the design and optimization of metallic anti-frosting SHSs by ultrafast laser processing.

## 1. Introduction

Condensation frosting occurs on the metal fins or tubes of cryogenic equipment, including air conditioning units and refrigeration systems [1]. Frosting can reportedly lead to a 30–57% decrease in the heating capacity of air-source heat pumps, and defrosting cycles can consume up to 12.9% of the total operating energy [2]. Moreover, frost accretion can reduce the lift of aircraft wings by more than 30%, thus endangering the lives of passengers [3]. Therefore, the development of metallic anti-frosting surfaces that delay frosting and reduce frost coverage is of practical significance [4]. To date, the construction of anti-frosting superhydrophobic surfaces (SHSs), from which condensed droplets can be self-removed via coalescence-induced jumping, has been validated as a favorable strategy because it can effectively inhibit frost formation, propagation, and accumulation [5,6,7].

Previous studies have shown that rationally designed nanostructures on SHSs are crucial for achieving the coalescence-induced self-removal of condensed microdroplets [8,9]. Accordingly, various nanostructures, such as nanopores [10,11], nanospikes [12,13], nanoblades [14], nanotubes [15], nanowires [16,17], and nanocones [18], have been successfully constructed to prepare metallic SHSs via chemical oxidation, electrochemical deposition, and laser irradiation [19]. However, these methods are applicable only to a few pure metals, such as aluminum, copper, and zinc, thereby hindering the development of metallic anti-frosting materials and further application of condensate self-removing surfaces in advanced engineering fields, such as electronics, fuel cell systems, and aerospace [20,21,22]. Recently, short and ultrashort pulsed laser processing techniques have demonstrated material adaptability for the fabrication of metallic anti-frosting surfaces. In particular, SHSs based on copper [23], titanium [24], aluminum alloys [25], and stainless steel [26,27] have been prepared, and their anti-frosting performance has been demonstrated. However, the condensate self-removal capability of such surfaces is yet to be analyzed in detail, and guidelines for parameter control for the fabrication of rationally structured SHSs via laser processing are yet to be established.

Therefore, this study uses ultrafast laser processing to facilely fabricate a series of nanostructured copper-zinc alloy surfaces with varied surface morphologies by adjusting the processing parameters [28,29]. The length scale of the surface nanofeatures was controllable down to the order of a hundred nanometers; this is favorable for achieving efficient self-propelled jumping of condensed droplets [21]. The condensation process and self-propelled droplet jumping events in samples with different morphologies were observed and analyzed to explore the relationship between the surface morphology and the condensate self-removal capability. The anti-frosting potential of surfaces with different water-removal capabilities was evaluated. Enhanced self-propelled droplet jumping through the suitable tuning of the laser processing parameters was found to play a decisive role in improving and optimizing the anti-frosting performance of SHSs. A guideline for laser processing parameterization was successfully established to fabricate metallic anti-frosting SHSs with optimal performance. Furthermore, the material adaptability of the ultrafast laser processing technique was validated by extending the methodology to aluminum, carbon steel, and titanium-aluminum-vanadium alloys. This study provides a valuable understanding of SHS structuring for anti-frosting and serves as a practical parameter framework for the fabrication of metallic anti-frosting SHSs via laser processing.

## 2. Materials and Methods

### 2.1. Materials

Samples of copper-zinc alloys (Appendix A), aluminum, carbon steel, and titanium-aluminum-vanadium alloys (each with dimensions of 75 × 75 × 0.5 mm^3^) were mechanically polished and cleaned ultrasonically using ethanol and deionized water prior to laser treatment.

### 2.2. Fabrication Procedure

A linearly polarized picosecond laser (Edgewave PX series, Würselen, Germany) with a wavelength, maximum pulse repetition frequency, and pulse width of 1064 nm, 20 MHz, and 12 ps was used to irradiate the as-prepared samples, respectively. The maximum output laser power was 70 W. A two-mirror galvanometer scanner (Scanlab, Puchheim, Germany) with an F-Theta objective lens (*f* = 170 mm) was used to focus and scan the laser beam in the x-y plane, and the focused diameter (*d*) of the Gaussian-profile laser beam at 1/e^2^ of its maximum intensity was ~50 μm. Laser processing experiments were performed in a cleanroom environment under normal laser beam incidence. The laser scanning path was designed to be line-by-line, and the interval between adjacent laser scanning paths (Figure 1a) was maintained at 20 μm. During surface structuring, the laser power varied between 0 and 70 W, and the pulse repetition frequency varied between 300 kHz and 2 MHz, whereas other laser parameters (e.g., scanning speed and scan number) were maintained as constant. Approximately 10 min was needed to fabricate one sample (75 × 75 × 0.5 mm^3^); increasing the laser power, laser spot diameter, scan interval, and scan speed and performing parallel processing increased the fabrication efficiency [30].

After laser irradiation, the samples were coated with a thin layer of 1H,1H,2H,2H-perfluorodecyltrimethoxysilane (FAS-17) via vapor deposition at 60 °C for 4 h.

### 2.3. Surface Characterization

The surface morphologies were analyzed using scanning electron microscopy (SEM; Zeiss scanning electron microscope, Zeiss, Jena, Germany). The surface composition measurements were analyzed using energy-dispersive X-ray spectrometry (EDS). Surface topology measurements were performed using white light interferometry (ZYGO NexView, Midlefield, CT, USA). Contact angle (CA) measurements were performed using a video-based optical CA measuring device (OCA 15 Plus from Data Physics Instruments, Filderstadt, Germany) and the sessile drop technique using water droplets of volume 5 μL; an average of three readings on different surface locations was used.

### 2.4. Condensation Experiments

Thermal paste was used to bond metal samples onto a vertically mounted cryostage (Linkam LTS420, Redhill, UK). The ambient relative humidity and temperature were 55.8 ± 5% and (23.4 ± 2.5) °C, respectively. The temperature of the cryostage was maintained at 1 °C. A mounted high-speed video camera (Phantom v7.3, Vision Research, Inc., Wayne, NJ, USA) with an optical lens was used to record the condensation process of each sample for 1 min at 10 fps. “Time zero” refers to 30 min after the onset of the condensation test.

### 2.5. Evaporation Experiments

To estimate the advancing CAs (ACAs) and receding CAs (RCAs) for condensed microdroplets, the prepared samples were horizontally bonded onto a cryostage with thermal paste. The ambient relative humidity and temperature were 45.7 ± 5% and 26.1 ± 2.5 °C, respectively. The temperature of the cryostage was maintained at 1 °C for 10 min. After 10 min, the cryostage temperature was increased to ambient temperature, and condensed droplets on the samples slowly evaporated away. The evaporation processes were recorded using a digital camera platform with a 10× magnification objective lens (OCA 15 Plus from Data Physics Instruments, Filderstadt, Germany), and the ACA and RCA were measured manually. The CAs were estimated by averaging the results from 3–4 evaporation tests with an uncertainty of approximately ±10° (Appendix A).

### 2.6. Frost Coverage Experiment

Thermal paste was used to bond the samples onto a vertically oriented Peltier device. The ambient relative humidity and temperature were 42.8 ± 5% and (22.5 ± 2.5) °C, respectively. The temperature of the Peltier device was maintained at (−7 ± 1.5) °C (supersaturation degree (*SSD*) = 2.45). The frosting processes of the samples were recorded using a digital camera (Canon EOS 60d, Canon, Tokyo, Japan), and a picture was captured every 10 min.

### 2.7. Frost Propagation Experiment

Thermal paste was used to bond samples onto the cryostage. The ambient relative humidity and temperature were 26.4 ± 5% and (20.7 ± 2.5) °C, respectively. The temperature of the cooling plate was maintained at (−10 ± 1.5) °C (*SSD* = 2.25), and the frosting propagation processes of the samples were recorded using a digital camera at 1 fps (Nikon DS-Ri1, Nikon, Tokyo, Japan).

Notably, the experimental conditions for the frosting propagation and frost coverage tests were different. First, two sets of frosting experiments were conducted under different environmental conditions because we were unable to precisely control the ambient conditions (e.g., temperature and humidity). Second, the *SSD* in the two sets of experiments was similar (for large samples, *SSD* = 2.25 and 2.45 in the frost propagation test and frost coverage test, respectively) despite the differences in the ambient temperature and relative humidity. More importantly, studies have reported that ice bridge formation between the frozen droplet and its neighboring liquid droplet depends predominantly on the ratio (*L*/*D*) of the distance between the frozen and neighboring evaporating droplets (*L*) and the size of the evaporating droplet (*D*), and not on the ambient conditions [31]. Further, because the probability of successful ice bridge formation between the droplets determines the overall frost propagation speed, even if the conditions for frosting tests are different, surfaces with a lower water accumulation amount (i.e., large *L*/*D*) should likely have a lower chance of successful ice bridge formation, and thus, a slower frost propagation speed. In contrast, if the water accumulation amount increases (i.e., smaller *L*/*D*), the chance of successful ice bridge formation between the droplets is higher; this corresponds to a higher overall frost propagation speed. For instance, at a low *SSD*, the surface-type S3-Cu exhibits the most effective condensate removal owing to its appropriate nanostructural morphology, thus resulting in the lowest amount of accumulated water and, in turn, the lowest frost propagation speed among all the surface types. When *SSD* increases, more condensed water forms on S3-Cu; nonetheless, it exhibits the lowest accumulated water amount (under no flooding condition) and lowest frost propagation speed compared with the other surface types because it still possesses the strongest water removal ability. Therefore, the differences in the ambient conditions between the frosting tests should not be an issue in relating the frost propagation speed to the eventual frost coverage of the prepared samples.

## 3. Results and Discussion

To determine the relationship between the laser processing parameters and nanostructural morphologies, a series of copper-zinc alloy (Cu) samples were fabricated using a single laser scan (Figure 1a; see details in the Section 2). During laser processing, the laser scanning interval was fixed at 20 µm and the scanning speed (*v*) at 1 m/s. Subsequently, the laser fluence (*f*) and laser pulse repetition frequency (*PRF*) were varied according to the parameters listed in Appendix A. Laser fluence is calculated as *f* = *4P*/(*PRF* × *πd^2^*), where *P* is the average laser input power (unit: W); *PRF* is the number of deposited laser pulses per unit time (unit: kHz); and *d* is the focused diameter of the Gaussian-profile laser beam at 1/e^2^ of its maximum intensity (unit: cm). After laser processing, the structured surfaces were further treated with FAS-17 to make them superhydrophobic. Herein, 25 different Cu samples were facilely prepared, and superhydrophobic Cu surfaces with an advancing water CA > 163° were produced (Figure 1b, Appendix A).

The 25 prepared samples were categorized into three types according to the morphology of the structured surfaces (Figure 1c–e). When the laser fluence was below the ablation threshold of the material, no damage occurred. When the laser fluence was just above the threshold, a small number of submicrometer structures were induced on the irradiated surface (Figure 1c). The heights of these structures were <2 μm (Appendix A). This surface type corresponded to S1-Cu. With an increase in the laser fluence, laser-induced periodic surface structures (LIPSS) with a periodicity of ~1 μm (corresponding to the wavelength of the laser beam) and a height of ~2 μm were formed (Figure 1d and Appendix A) [32]. Additionally, ultrafast laser processing above the ablation threshold of the irradiated surface led to the ejection and redeposition of nanoscale particulates back onto the surface [33]. The resulting surface type is comprised of LIPSS with cellular-like nanofeatures having a spacing of a few hundred nanometers, denoted as S2-Cu. As the laser fluence was further increased to well above the ablation threshold, the irradiated laser pulses on the Cu surface induced a significant amount of plasma formation that consisted primarily of oxidized copper and zinc nanoparticles (Appendix A) [34]. A fraction of these nanoparticles gained sufficiently high initial kinetic energy to be ejected from the Cu surface, and they eventually dissipated into the free space. The remaining nanoparticles lost their initial kinetic energy and were redeposited onto the Cu surface owing to gravity and ambient air pressure to form abundant nanoparticle clusters. The heights of the nanofeatures increased to more than 4 μm, and the spacing decreased to ~100 nm (Figure 1e and Appendix A); this surface type was denoted as S3-Cu. Under a similar laser fluence, as the *PRF* increased from 300 to 2000 kHz, more laser pulses were injected onto a specific surface spot. This led to increased material removal and a larger number of deposited nanoparticles on the irradiated surface.

For solid surfaces having the same chemical nature, the wetting properties depend on their respective morphologies [35]. The three types of Cu surfaces exhibited considerable differences in adhesion for macroscale droplets. Figure 2a1 shows the pinning of deposited millimeter droplets (5 μL) to S1-Cu with an advancing contact angle (ACA) of 145.5 ± 1.5°. The droplet could not slide on the substrate because of strong liquid-solid adhesion resulting from complete wetting in the Wenzel state. The droplets deposited on S2-Cu (Figure 2b1) had an ACA of 146.8 ± 1.6° and a RCA of 134.8 ± 3.7°. The droplet could slide on the substrate, albeit with a tilting angle exceeding 30°. This indicated lower liquid-solid adhesion than that in the case of S1-Cu; nonetheless, the adhesion was sufficiently strong to induce sufficient contact angle hysteresis. The microliter droplets deposited on S3-Cu (Figure 2c1) easily rolled off at a tilting angle of 2°, thus indicating excellent superhydrophobicity toward macroscopic water droplets. Appendix A summarizes the contact angle analysis for macroscale droplets on Cu surfaces.

To characterize the adhesion interaction of the surfaces with condensed microdroplets and to evaluate their anti-frosting potential, the wetting properties of condensed microdroplets should be assessed for the three types of surfaces [36]. Notably, a surface that is nonwetting to macroscale droplets is not necessarily superhydrophobic to condensed microdroplets, owing to the possible wetting of surface textures [37]. Therefore, previously reported conclusions from adhesion force analysis for macroscale droplets may not accurately determine the adhesion for condensed microdroplets [38]. The droplet evaporation test is a better method for evaluating the wetting property of a microdroplet. Quantitative measurements of the contact angles of condensed microdroplets were performed in the evaporation tests (Appendix A; see details in the Section 2). Appendix A revealed that S3-Cu exhibited superhydrophobicity to condensed microdroplets as they were nearly spherical in shape. In addition, the ACAs of S2-Cu (165 ± 5°) and S3-Cu (155 ± 5°) were similar; however, the RCA of S2-Cu (90 ± 10°) was significantly lower than that of S3-Cu (135 ± 10°). These results indicated that S2-Cu was possibly in a partial-wetting state with a higher solid-liquid adhesion compared with S3-Cu (Appendix A) [39]. For S1-Cu (Appendix A), condensed droplets completely wetted the surface in the Wenzel state (ACA = 130 ± 5°, RCA = 30 ± 10°), thus resulting in very high solid-liquid adhesion and preventing any coalescence-induced droplet jumping. However, accurately measuring the contact angle of a sessile macroscopic droplet on a superhydrophobic surface remains difficult, and doing so for condensed microdroplets (<50 μm) is even more difficult [40]. Currently, the contact angles of microdroplets are estimated through specialized techniques such as piezoelectric microgoniometry [41], laser scanning confocal microscopy [42], and other optical techniques [43]. However, these approaches are rather expensive and unavailable in this study. Herein, the wetting properties of the prepared samples were evaluated according to the studies by Wen et al., wherein the droplet-surface adhesion is indirectly characterized with respect to their coalescing mobility (e.g., high droplet jumping frequency corresponds to less surface wetting, and vice versa) [44].

To observe the droplet coalescing behaviors, condensation experiments were performed using the setup shown in Appendix A (see details in the Section 2). Figure 2a2,a3 show that on S1-Cu, no coalescence-induced droplet jumping event was observed during a 10-s interval (0 min^−1^⋅cm^−2^), and a considerable amount of condensed water accumulated on the surface (~2.71 mg/cm^2^) over 30 min of the condensation experiment. Water accumulation was determined with the same method as in the previous study [45,46]. These observations indicated that the condensed microdroplets completely wetted the surface structures that were constructed with a low laser fluence. For S2-Cu, as the laser fluence increased, the resulting surface comprised LIPSS with cellular-like submicrometer-scale structures. Very few droplet jumping events were observed upon coalescence (~350 min^−1^⋅cm^−2^), and most coalesced droplets failed to jump (Video S1). Nevertheless, the water accumulation amount for S2-Cu (1.44 mg/cm^2^) was lower than that for S1-Cu (Figure 2b2,b3). When the laser fluence was well above the ablation threshold, the surface features comprised dense redeposited nanoparticles. For surface type S3-Cu, the amount of accumulated water (0.26 mg/cm^2^) was significantly reduced compared to those for S1-Cu and S2-Cu over 30 min of the condensation experiment. This was attributed to efficient jumping (~5600 min^−1^⋅cm^−2^) and continuous removal of condensed droplets (Figure 2c2,c3, Appendix A).

The droplet jumping frequency may be determined by the surface morphology. Solid-liquid interaction on rough surfaces can be categorized into the Wenzel (W), Cassie–Baxter (CB), and the recently proposed partial-wetting (PW) states (Figure 3a–c) [47]. Reportedly, the wetting property of a condensed microdroplet depends on the tip area (*t*), center-to-center pitch (*l*), and height (*h*) of the nanofeatures [9,48]. Crucially, *h* must exceed a critical value (e.g., *h* > *l*) because nanostructures are completely wetted by the condensed water droplet when *h* is small owing to the disruption in the Capillary-Laplace balance [8]. Additionally, a large *h* leads to less vapor diffusion into the nanostructure, where structure wetting becomes more difficult and droplet pinning is less likely to occur [49]. Moreover, adhesion at the solid-liquid interface must be minimized for effective droplet jumping [50,51]. The solid-liquid interaction strongly depends on the nanostructural topology of the surface. Particularly, the tip area (*t*) and spacing between the nanofeatures (*l*) determine the jumping capability of coalesced droplets [9]. Generally, a smaller *t* is beneficial for droplet jumping because the solid-liquid interface area is reduced, and a larger *l* is preferred for the same reason. However, if *l* increases too much, moisture penetration becomes inevitable. Therefore, *l* should be restricted to the submicrometer scale (e.g., ~100 nm) [8].

In this study, when the laser fluence was just above the ablation threshold, surface features were deficient and *h* was small, thus resulting in the nonjumping of condensates after coalescence because of the complete wetting of surface structures (S1-Cu) (Figure 3a). When the laser fluence increased to 0.51 J/cm^2^, the nanostructures consisted primarily of cellular-like structures atop LIPSS (S2-Cu). The adhesion at the solid-liquid interface decreased with respect to S1-Cu; however, it still posed as a barrier to the self-removal of condensed droplets because they were unable to efficiently jump off the surface through coalescence. This can be attributed to both the severe partial-wetting of nanostructures due to a small *h* and the increased adhesion from a relatively large *t* (Figure 3b). When the laser fluence was further increased to more than 1.76 J/cm^2^, an abundance of redeposited nanoparticle-constructed structures (S3-Cu) was observed, wherein condensed droplets could grow and navigate out from the nanofeatures and suspend in the CB state [52]. Consequently, a layer of air could be effectively trapped between surface structures and condensed droplets to prevent the wetting of nanostructures (Figure 3c). The adhesion at the solid-liquid interface was significantly decreased when *t* became smaller. S3-Cu promoted efficient removal of nonwetting condensates through coalesced jumping (Figure 3c). Moreover, a theoretical analysis was provided to further demonstrate the effect of the droplet wetting properties on the self-propelled droplet jumping capabilities (Appendix A, Section 3, and Appendix A).

To instruct the design and fabrication of condensate self-removing surfaces for anti-frosting using ultrafast laser processing, a quick optimization method was proposed to obtain the process window for achieving optimal anti-frosting performance. The droplet jumping frequency during a 1-min condensation test was evaluated for all 25 facilely prepared Cu samples, and the results were related to the laser processing parameters through a phase map (Figure 3d). Notably, the laser processing parameters and droplet jumping frequency were selected as parameters for optimizing the anti-frosting performance instead of the surface morphology and surface wetting property. This is because the surface nanostructures can only be rationally defined when they are regular (e.g., an array of nanocones) [9]. However, for the 25 Cu samples, the nanostructures were irregular (i.e., random nanoparticle clusters), thus rendering difficulty in quantifying the exact nanostructural parameters (e.g., height, tip area, and center to center pitch). In terms of wetting properties, accurately measuring the contact angles of condensed microdroplets without specialized instruments is challenging, and the surface wetting properties are determined by the jumping frequency of coalesced droplets. Therefore, from the viewpoint of surface design, considering two parameters that are easy to obtain (laser processing parameters and droplet jumping frequency) is feasible for instructing the design of an optimal anti-frosting surface, except when using parameters that are difficult to quantify (e.g., surface geometrical dimensions, contact angles).

From the phase map, the region with the most frequent droplet jumping events had a laser fluence of >5 J/cm^2^ (Figure 3d), which corresponded to a lower amount of accumulated water (Figure 3e). When *PRF* was less than 1500 kHz, it did not drastically influence the condensate self-removal capability of Cu samples. Typically, a higher *PRF* results in more ablated and redeposited nanoparticles, which is beneficial for droplet jumping. However, when the *PRF* reached 2000 kHz, severe melting of the Cu surface and the formation of microholes occurred because of the “accumulation effect” (Appendix A). This led to a greater amount of condensed water owing to the larger effective condensing area and resulted in increased water accumulation [53]. A general design guideline for fabricating a surface with high droplet removal capability and minimized water accumulation involves using sufficiently high laser fluence and *PRF* to produce abundant nanoparticle-constructed structures. However, the severe surface melting that results from excessive heat input from the exorbitantly high laser fluence and *PRF* must be avoided. By carefully tuning the surface morphology with optimized laser processing parameters (such as S3-Cu), the droplet jumping frequency could exceed 5676 min^−1^⋅cm^−2^, and its corresponding accumulated water amount on the surface was reduced to 0.26 mg/cm^2^ or less than one-fifth of that of S1-Cu over 30 min of the condensation experiments. Such an optimized surface should be favorable for anti-frosting (Figure 3d–e) [54].

A surface can exhibit effective anti-frosting performance in a condensing environment if the frost delay time is prolonged and frost propagation is inhibited. Generally, small-sized condensed droplets on a surface are preferable to reduce the probability of heterogeneous nucleation at the solid-liquid interface. This is positively correlated to the solid-liquid contact area *A_solid-liquid_ = φπR^2^sinθ*, where *φ* is the solid fraction; *R*, the radius of the accumulated droplet; and *θ*, the static contact angle of the droplet [55]. Because *φ* and *θ* are constant values for condensed droplets on a nanostructured superhydrophobic surface, the freezing delay time of a supercooled droplet depends on its radius [56,57]. Appendix A shows that the amount of large accumulated droplets with diameters > 100 μm was less than 30 cm^−2^ for S3-Cu over 30 min of the condensation experiment, as compared to ~500 and ~900 cm^−2^ for S2-Cu and S1-Cu, respectively. Therefore, condensed droplets on S3-Cu should exhibit a longer freezing delay time.

On a sufficiently large surface where the influence of the edge effect can be neglected, once a few condensates randomly freeze into ice after a delay, condensation frosting is dominated by frost propagation [58,59]. Particularly, an interdroplet ice bridge forms between a frozen droplet and its neighboring liquid droplets owing to the presence of a vapor pressure gradient, which triggers a chain reaction of freezing events [60]. Boreyko et al. studied the mass transfer from an evaporating liquid droplet to a neighboring frozen droplet and attributed successful interdroplet ice bridge formation to a simple relationship: *L*/*D*, where *L* is the interdroplet distance between a frozen droplet and its neighboring unfrozen droplet, and *D* is the diameter of the evaporating droplet [61]. Crucially, a large *L*/*D* is desirable to prevent ice bridge formation via the complete evaporation of a liquid droplet before it connects with a frozen droplet to form a so-called dry zone. Herein, using similar reasoning, the droplet size distribution on the samples after 30 min of the condensation experiment was statistically analyzed (Appendix A). The average droplet diameters (*D_avg_*) were 26.33, 20.22, and 17.93 μm for S1-Cu, S2-Cu, and S3-Cu, respectively. Moreover, the average distance (*L_avg_*) between condensed droplets and their closest neighbors was measured. Appendix A shows that for the surface with the most effective droplet jumping (S3-Cu), more than 30% of the droplets had a *L_avg_*/*D_avg_* > 1. In contrast, only 21.9 and 15.5% of the droplets on S2-Cu and S1-Cu, respectively, had *L_avg_*/*D_avg_* > 1, which indicated that the frost propagation speed of S3-Cu should be significantly lower than that of S1-Cu and S2-Cu [54].

To verify the practicality of the design methodology for ultrafast laser constructed anti-frosting surfaces, frosting tests were conducted on large samples using the setup illustrated in Appendix A. To mitigate the influence of the edge effect, large samples (75 × 75 × 0.5 mm^3^) were prepared and bonded onto a vertically oriented Peltier device with thermal paste. The ambient relative humidity was 42.8 ± 5%, and the temperature was 22.5 ± 2.5 °C. The cooling plate was maintained at a temperature of −7 ± 1.5 °C (*SSD* = 2.45), and a photograph was captured every 10 min during the frosting experiments. Evidently, frost first appeared on the bare Cu plate after 5 min, and the surface was entirely covered with frost in 10 min due to the ease of ice nucleation and frost propagation (Figure 4). For the optimized S3-Cu, frost first appeared randomly on the surface after 60 min, and the frost coverage remained below 70% even after 120 min during a frosting experiment. Next, defrosting tests were conducted on Cu-S3. When the substrate temperature was increased above 0 °C, the entire frost layer quickly peeled off the surface within 50 s without leaving any residual droplets, thus indicating that it did not penetrate the surface nanotexture (Appendix A) [62].

The effective anti-frosting performance of S3-Cu can be more closely examined by comparing its frost propagation process with that of other samples (Figure 5, see details in Experimental). Importantly, owing to efficient condensate removal on S3-Cu, sustained dry zone formation was observed around the propagating frost front (Figure 5d). In contrast, frost propagated on other samples through the formation of interdroplet ice bridges, as in previous studies without proper surface optimization (Figure 5a–c) [63,64,65]. Consequently, the frost propagation speed could be reduced by a factor of 48 on S3-Cu (0.47 μm/s) compared with bare Cu plate (22.47 μm/s). This finding demonstrates the importance of tuning the surface morphology by adjusting the laser processing parameters for high water removal capability to achieve superior anti-frosting performance.

The ultrafast laser processing technique has no apparent material dependence, and the laser processing strategy adopted for constructing anti-frosting Cu should be applicable to the fabrication and tuning of other metal and metal-alloy surfaces. Figure 6a–c demonstrate that surfaces with appropriate nanostructures were successfully formed on aluminum (Al), carbon steel (steel), and titanium alloy (Ti) surfaces, and superhydrophobicity was achieved for both macro- and microscale droplets (Appendix A). The surface nanostructures produced on Al closely resembled those produced on Cu. Appendix A show that the effects of laser processing parameters on the surface morphology and condensate self-removal capability for Al followed similar trends to those of Cu. Further, the design process window for Al was comparable to that for Cu (Figure 6a2).

However, the requirements for effective droplet jumping are much more stringent for steel and Ti, and laser processing parameters must be chosen carefully to achieve the desired condensate self-removal capability. Appendix A show that a low laser fluence did not provide sufficient redeposited nanoparticle-constructed structures for condensed droplets to remain in the CB state, and no jumping event occurred (as in the cases of Cu and Al). Additionally, high laser fluence and high *PRF* resulted in severe melting of the surfaces owing to their lower melting threshold. Consequently, the nanostructures were destroyed, and the surfaces became flat. This led to high adhesion between microdroplets and surfaces [66]. Therefore, the process window for obtaining high condensate self-removal capability and a low amount of accumulated water on steel and Ti is noticeably narrower than that for Cu and Al (Figure 6b2–c2).

Figure 7a,b show the anti-frosting performance of the optimized Al (S3-Al), steel (S3-Steel), and Ti (S3-Ti) surfaces. Frost first appeared on all bare plates after 5 min, and the entire surface was covered within 10 min. In contrast, the frost coverage after 120 min during the frosting test for S3-Al, S3-steel, and S3-Ti was only 83, 76, and 81%, respectively. All S3-type surfaces distinctly exhibited significantly prolonged frost delay and reduced frost coverage over time in comparison to their bare counterparts. This is a result of the efficient removal of condensed microdroplets from the surfaces, and it demonstrates the effectiveness of our design, optimization, and fabrication strategy.

## 4. Conclusions

In this study, 25 copper-zinc alloy (Cu) surfaces were prepared via ultrafast laser processing. After hydrophobic modification, surfaces having an abundance of deposited nanoparticles exhibited effective condensate self-removal through coalescence. Furthermore, the influence of the laser-induced surface morphology on the condensate self-removal capability was investigated in detail, and a quick optimization method that can easily pinpoint the process window for the efficient self-removal of condensed microdroplets was developed. Crucially, by carefully tuning the surface morphology by adjusting the laser processing parameters, the condensate self-removal capability could be enhanced remarkably, and the corresponding surface type exhibited superior anti-frosting performance, with less than 70% frost coverage observed after 120 min during the frosting experiment (*SSD* = 2.45) owing to a significant reduction in the frost propagation speed (<1 μm/s). Finally, the simple ultrafast laser fabrication and optimization method was extended to other metallic materials for anti-frosting, and its excellent material adaptability was demonstrated. This study provides valuable insights for the design, fabrication, and optimization of metallic anti-frosting SHSs using ultrafast laser surface structuring.

## Figures and Tables

**Figure 1 nanomaterials-12-03655-f001:**
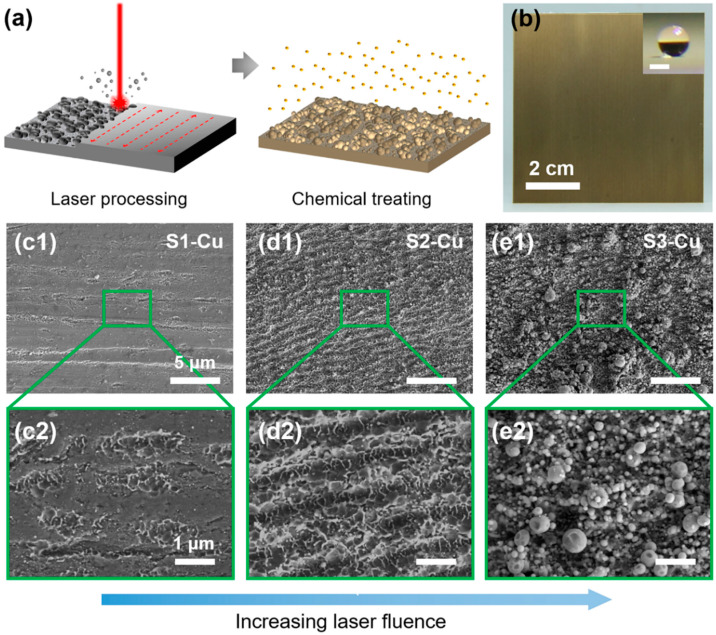
Preparation of condensate self-removing surfaces. (**a**) Schematic showing the fabrication process of the samples via ultrafast laser scanning. (**b**) Digital photographs of the resulting superhydrophobic surface (75 × 75 mm^2^); scale bar in the inset is 1, in which (**c**–**e**) SEM images of the representative copper-zinc alloy (Cu) surface types with distinct surface morphologies for (**c1**,**c2**) S1-Cu, (**d1**,**d2**) S2-Cu, and (**e1**,**e2**) S3-Cu.

**Figure 2 nanomaterials-12-03655-f002:**
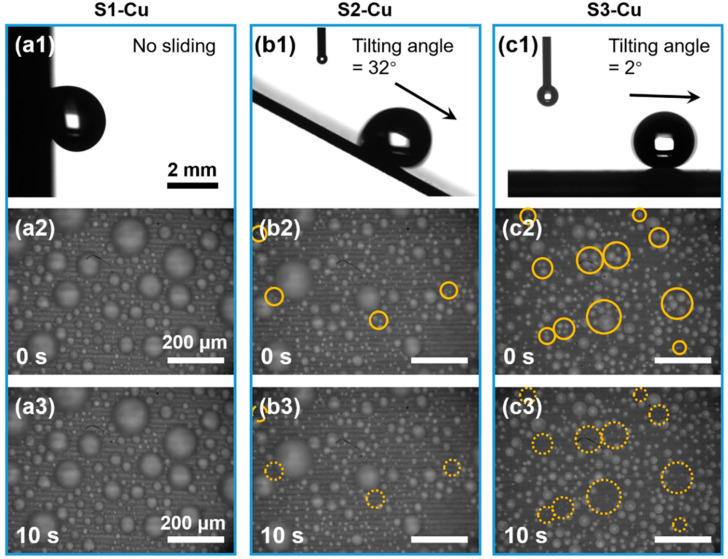
Wetting properties of sessile macroscale and condensed microscale droplets on the Cu samples. (**a1**–**c1**) Sliding behavior of deposited macroscopic droplets on S1-Cu, S2-Cu, and S3-Cu, respectively. (**a2**,**a3**) S1-Cu, condensed microdroplets fail to jump through coalescence, which leads to a high amount of accumulated water. (**b2**,**b3**) S2-Cu, condensed microdroplets jump with low efficiency. (**c2**,**c3**) S3-Cu, condensed microdroplets jump efficiently through coalescence, where they can be promptly removed from the surface before growing large. The images are taken 30 min after the onset of condensing, and the droplet jumping frequency on the surfaces is illustrated over a 10-s interval.

**Figure 3 nanomaterials-12-03655-f003:**
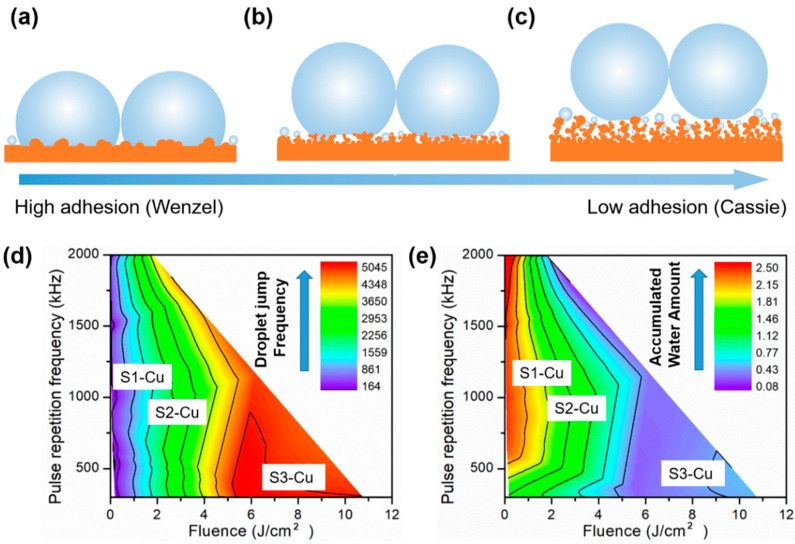
Effects of laser processing parameters on surface morphology and wetting properties of condensed droplets. (**a**) Schematic showing Wenzel droplets coalescing with no occurrences of jumping. (**b**) Schematic showing coalescence of partial-wetting droplets; inefficient droplet jumping results from a lack of surface nanostructure, roughness, and structural height. (**c**) Schematic showing the coalescence of CB droplets, which leads to efficient droplet jumping. (**d**) Phase map relating laser processing parameters to the number of droplet jumping events for Cu. The red and purple regions represent the most frequent and least frequent (or nonjumping) events, respectively. (**e**) Phase map relating laser processing parameters to the amount of accumulated water for Cu. The red and purple regions represent the highest and lowest amounts of accumulated water, respectively. Unit for the number of jumped droplets: min^−1^·cm^−2^; unit for accumulated water over 30 min of the condensation experiment: mg/cm^2^.

**Figure 4 nanomaterials-12-03655-f004:**
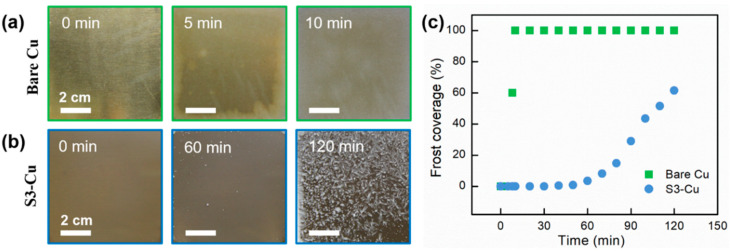
Anti-frosting performance of Cu samples. (**a**) Frost completely covers the entire bare Cu surface after 10 min. (**b**) Frost first appears after 60 min during a frosting test on the optimized S3-Cu sample, and the coverage remains <70% after 120 min under high *SSD* (2.45). (**c**) Frost coverage comparison between bare Cu and S3-Cu.

**Figure 5 nanomaterials-12-03655-f005:**
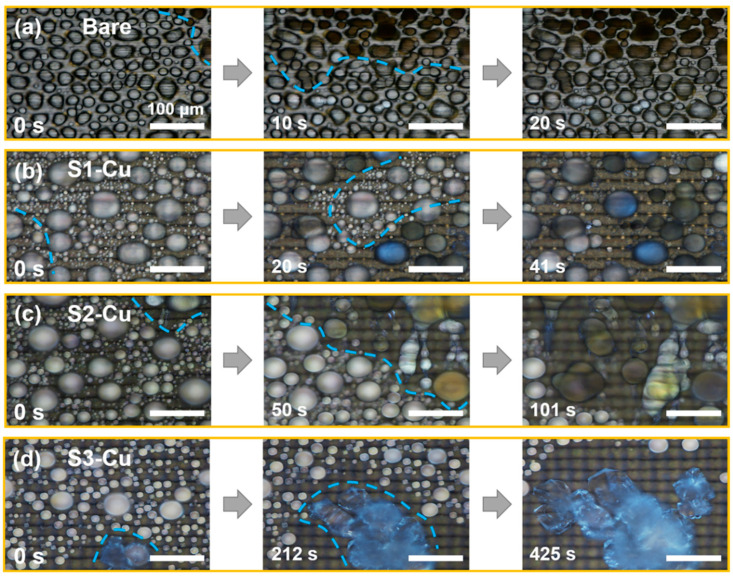
Frost propagation on the prepared surfaces. (**a**) Bare Cu with a frost propagation speed of 22.47 μm/s. (**b**) S1-Cu with frost propagation speed of 10.96 μm/s. (**c**) S2-Cu with a frost propagation speed of 4.45 μm/s. (**d**) S3-Cu with frost propagation speed of 0.47 μm/s; notably, dry zone formation dominates the frosting process of S3-Cu. The ambient relative humidity and temperature are 26.4 ± 5% and (20.7 ± 2.5) °C, respectively. The cryostage is maintained at (−10 ± 1.5) °C (*SSD* = 2.25; the frosting propagation processes are recorded using a digital camera at 1 fps (Nikon DS-Ri1).

**Figure 6 nanomaterials-12-03655-f006:**
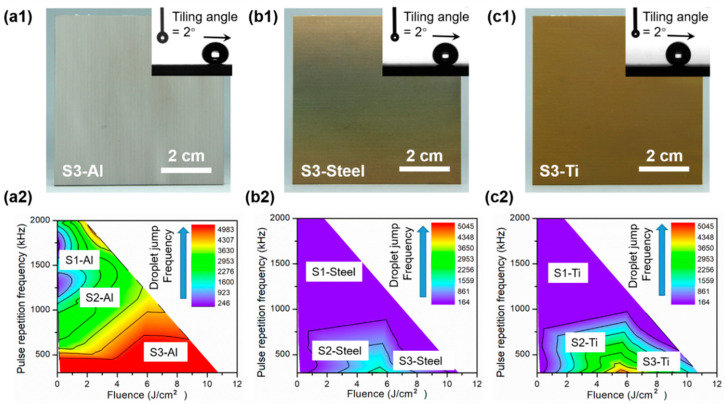
Ultrafast laser-constructed SHSs and condensed microdroplet behavior phase maps for other metal and metal alloy surfaces. (**a1**–**c1**) Large S3-type Al, steel, and Ti surfaces. (**a2**–**c2**) Phase maps relating laser processing parameters to the droplet jumping frequency for Al, steel, and Ti. Units for the number of removed droplets: cm^−2^·min^−1^.

**Figure 7 nanomaterials-12-03655-f007:**
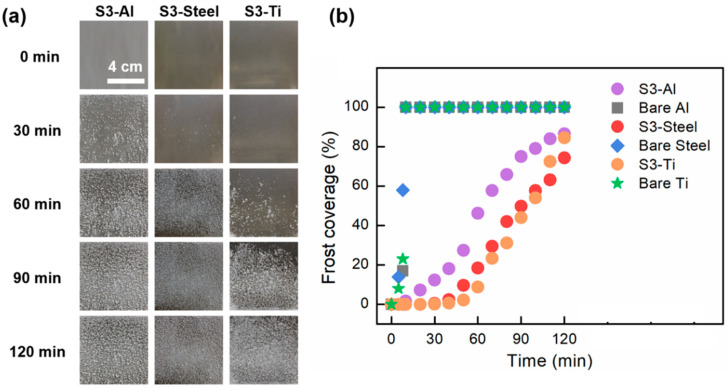
Frost test (**a**) and frost coverage (**b**) with respect to time for S3-Al, S3-steel, and S3-Ti surfaces and their respective bare counterparts.

## Data Availability

Data will be made available on request.

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
