# Peer review of "Fabrication of Metallic Superhydrophobic Surfaces with Tunable Condensate Self-Removal Capability and Excellent Anti-Frosting Performance"

_nanomaterials, 2022, doi:10.3390/nano12203655_

Round 1

Reviewer 1 Report

Contact angle as a function of V should be shown

22.5 ± 2.5 °C should be writen as (22.5 ± 2.5) °C

Fig.4. the noncontinous increase for bare Cu at low time should be expalined, similary Fig. 7. How big is the standard error?

Author Response

Thank you for your comments concerning our manuscript. We have studied comments carefully and have made the correction. We hope it meets with approval. The main corrections in the paper and the responds to the editor’s comments are listed below point by point as follows:

Response to Reviewer 1 Comments

Comment 1:

Contact angle as a function of V should be shown.

Response 1:

Thanks to the reviewer. The wetting properties of the laser textured surfaces are detected via contact angle measuring. It is mentioned that adhesion behaviors of the surfaces rationally evaluate its anti-frosting potential during condensation. Although the substrates experienced laser processing became superhydrophobic surface after chemical modification, different wetting states were observed. Thus, it is insufficient to determine wetting properties only by contact angle, but advancing contact angles and receding contact angles were compensated. With this stand, contact angle is employed for detecting the surface whether or not a superhydrophobic surface, it was a series of qualitative experiments. Additionally, volume of the droplets involved in contact angle testing was commonly known at definite volume like 5 μL. Several references can back up this selection of volume, ‘Hauschwitz, P., et al. (2020). "Hydrophilic to ultrahydrophobic transition of Al 7075 by affordable ns fiber laser and vacuum processing." Applied Surface Science 505: 144523.’ and ‘Pan, R., et al. (2021). "Triple-Scale Superhydrophobic Surface with Excellent Anti-Icing and Icephobic Performance via Ultrafast Laser Hybrid Fabrication." ACS Applied Materials & Interfaces 13(1): 1743-1753.’. Obviously, contact angle is as a function of volume. It goes larger when volume is bigger until a saturated contact. But, in experimental study an enough and changeless volume is preferred, not a variable value. Moreover, rare article takes account of the relationship between volume and contact angle, besides the paper for the physical insights in three-phase contact system. This is out of the scope of our work.

In conclusion, the volume is selected by a common sense, and the relationship with contact angle is not necessary in this kind of research. We preferred 5 μL to detect the wetting properties in our work.

Comment 2:

22.5 ± 2.5 °C should be writen as (22.5 ± 2.5) °C.

Response 2:

Thanks for the suggestion. I have rewritten all the temperature value in the right form.

Comment 3:

Fig.4. the noncontinous increase for bare Cu at low time should be expalined.

Response 3:

Thanks for the suggestion. Noncontinuous increase of the condensation is down to the ease of ice nucleation and frost propagation. The bare Cu surface enables a faster nucleation and propagation of condensation than the laser processed surfaces. The bare surface is frost covered over 60% area after 5 min, and then entirely covered in 10 min. In comparison of the bare surface, laser textured surface introduced a great delay on frost covering, and created barriers to prevent the penetration of condensation. On the other hand, due to the time dependent evolution of the frost coverage on bare Cu and S3-Cu, less data was needed to specify the frost level on bare surface than the textured surface. As depicted in the dots plots, Figure 4 (b), sampling frequency on bare surface is one shot per 10 minutes, on the textured surface is nonuniformed.

In conclusion, the noncontinuous increase of the frost coverage on bare Cu surface is due to the noncontinuous photograph sampling, and also attribute to its natural mechanism of ease nucleation and propagation.

The according content is revised and list below.

Evidently, frost first appeared on the bare Cu plate after 5 min, and the surface was entirely covered with frost in 10 min due to ease of ice nucleation and frost propagation (Figure 4).

In contrast, frost propagated on other samples through the formation of interdroplet ice bridges, as in previous studies without proper surface optimization (Figure 5(a)–(c)) [54-56]. Consequently, the frost propagation speed could be reduced by a factor of 48 on S3-Cu (0.47 μm/s) compared with bare Cu plate (22.47 μm/s).

Comment 4:

Fig. 7. How big is the standard error?

Response 4:

Thanks to the reviewer. The experiment depicted in Figure 7 gives distinct result on the anti-frosting performance of different materials. The bare surfaces were attached as reference. This work aims to introduce a universal method to optimize the parameters in laser modified anti-frost surface on steel surface, and extend this to other different metallic materials. Thus, we did not take account with the standard error to quantitatively evaluate the performance at present stage. Moreover, based on our research route, the more accurate model of the presented method will be executed once the study for practical application on a definite material is going to start. It is sufficient to support our study about the extended use of the universal method to other materials based on the result shown in Figure 7. Several papers performed the similar research through comparative experiments without error bar also. The papers are below.

Competing Effects between Condensation and Self-Removal of Water Droplets Determine Antifrosting Performance of Superhydrophobic Surfaces, 2020.

Liquid-Infused Nanostructured Surfaces with Extreme Anti-Ice and Anti-Frost Performance, 2012.

Anti-frost coatings containing carbon nanotube composite with reliable thermal cyclic property, 2014.

We tried our best to improve the manuscript and made some changes in the manuscript. These changes will not influence the content and framework of the paper.

Once again, thank you very much for your comments.

Best regards,

Jianguo He

Reviewer 2 Report

The paper "Fabrication of Metallic Superhydrophobic Surfaces with Tuna-ble Condensate Self-Removal Capability and Excellent Anti-frosting Performance" is carefully prepared manuscript devoted to the important topic. The results are novel and definitely interesting.

Remarks.

1. I suggest that an accurate interpretation of the reported results calls for the calculation of the relevant dimensionless numbers, describing the process, namely the Bond number, the Reynolds number  and the Peclet number, see: 

Starostin, A., Strelnikov, V., Valtsifer, V. et al. Robust icephobic coating based on the spiky fluorinated Al2O3 particles. Sci Rep 11, 5394 (2021).

I think that the calculation of these numbers will help to understand the processes occurring under frosting.

2. Explicit calculation of the hierarchy of the characteristic time scales, related to the physical processes, involved into frosting and de-frosting will be also helpful (see the aforementioned  paper, please).

3. The physical mechanism of frosting should be discussed in more detail.

Author Response

Thank you for your comments concerning our manuscript. We have studied comments carefully and have made the correction. We hope it meets with approval. The main corrections in the paper and the responds to the editor’s comments are listed below point by point as follows:

Response to Reviewer 2 Comments

Comment 1:

I suggest that an accurate interpretation of the reported results calls for the calculation of the relevant dimensionless numbers, describing the process, namely the Bond number, the Reynolds number and the Peclet number,

see: Starostin, A., Strelnikov, V., Valtsifer, V. et al. Robust icephobic coating based on the spiky fluorinated Al2O3 particles. Sci Rep 11, 5394 (2021).

I think that the calculation of these numbers will help to understand the processes occurring under frosting.

Response 1:

Thanks to the reviewer. The Bond number denotes the ratio of gravitational force to the surface tension force. A small Bond number is preferred because liquid droplets are more likely to be suspended on the structures by the capillary force provided by the structures. The Peclet number is to estimate the relationship between thermal advection and thermal diffusivity. These dimensionless numbers give detailed elucidation on the condensation process from heterogeneous nucleation, growth and departure of the liquid droplet. Differently, direct research of jumping event was presented to determine the effectiveness of different laser parameters in our work. We do not need to involve the procedure of detailed droplet condensation process, since the guideline of parameters selection for optimizing the anti-frosting performance is immediate and efficient. Moreover, experimental investigation on the time-dependent frosting coverage and propagation velocity were performed to examine the textured surface. In general, we are going to find a practically universal methodology for laser-processed functionalization on metallic surfaces. What is more, for a better insight of the droplet jumping due to different morphology-induced wetting state, a surface energy-based analysis was carried out in the S3 supplementary materials. This section discusses the kinetic energy responsible for off-surface motion of the coalescence droplet, and gives quantitative evaluations on the relationships of droplet jumping velocity and droplet radius for wetting states of Cassie-Baxter, partial wetting and Wenzel. The analysis involves surface roughness factor, dimensional parameters and surface tension. These relate the laser parameters, surface morphology and droplets jumping capability. We think the energy-based analysis helps a better and more direct understand for morphology-induced surface anti-frosting than dimensionless analysis.

Accordingly, some revisions are made and some literatures about dimensionless analysis are added to the reference.

Comment 2:

Explicit calculation of the hierarchy of the characteristic time scales, related to the physical processes, involved into frosting and de-frosting will be also helpful (see the aforementioned paper, please).

Response 2:

Thanks to the reviewer. As content demonstrated in reply 1 to comment 1, more explicit calculation of frosting and de-frosting process is redundant in the scope of this work. The literatures involving the mechanism and calculation are helpful for the phenomena explanation in our work, and these published articles had been cited in our manuscript.

Comment 3:

The physical mechanism of frosting should be discussed in more detail.

Response 3:

Thanks to the reviewer. This study focuses on the fabrication of functional surface and its optimization method. Condensation frosting mechanism could be helpful in understanding the anti-frosting of laser-processed surface. However, as the reply 1 to comment 1 demonstrates that the mechanism is necessary but not the main body of this research. The frequency of droplet jumping events, time-dependent coverage and propagation velocity were performed to back up the topic. It is sufficient and systematical for this work to propel the application of this technology in more metallic materials.

We tried our best to improve the manuscript and made some changes in the manuscript. These changes will not influence the content and framework of the paper.

Once again, thank you very much for your comments.

Best regards,

Jianguo He

Reviewer 3 Report

A very interesting studies combing laser irradiation of surfaces, wettability and anti-icing properties of materials.

I've no major remarks with respect to this work. I recommend its publication provided some references as discussed and that the originality of this work is highlighted with respect to them.

https://doi.org/10.1016/j.apsusc.2016.01.019

https://doi.org/10.1038/s41598-019-49615-x

https://doi.org/10.1088/2631-8695/abf35f

Author Response

Thank you for your comments concerning our manuscript. We have studied comments carefully and have made the correction. We hope it meets with approval. The main corrections in the paper and the responds to the editor’s comments are listed below point by point as follows:

Response to Reviewer 3 Comments

Comment 1:

A very interesting studies combing laser irradiation of surfaces, wettability and anti-icing properties of materials. I've no major remarks with respect to this work. I recommend its publication provided some references as discussed and that the originality of this work is highlighted with respect to them.

https://doi.org/10.1016/j.apsusc.2016.01.019

https://doi.org/10.1038/s41598-019-49615-x

https://doi.org/10.1088/2631-8695/abf35f.

Response 1:

Thanks to the reviewer for the recommendation and very nice suggestion. The provided articles are added to the references and given some discussions accordingly.

Once again, thank you very much for your comments.

Best regards,

Jianguo He

Reviewer 4 Report

The authors present a study considering a laser-based technique for optimizing the surface structure in order to achieve the best anti-frosting performance. The paper is relatively well-written, but there are several very important concerns that might hinder its publication. 

1) Using laser beam for modifying the surface morphology is not new and in fact, it is quite a well-known technique (https://www.mdpi.com/1996-1944/13/24/5692 ; https://www.mdpi.com/2079-6412/11/1/58 - many more can be found in the scientific databases). From that point-of-view, this research is an incremental scientific advance. 

2) The authors claim that there is a lack of literature sources dealing with optimization of the laser parameters for tuning the icephobic properties of the surface, which is also not true (https://www.mdpi.com/2079-4991/11/1/135 ; https://www.scientific.net/KEM.926.1643 - and many more in the scientific databases). Based on 1) and 2), the reviewer strongly recommends the authors to deeply examine the current state-of-the-art, cite the mentioned and other relevant literature, and then clearly outline what the main novelty of their research is. 

3) The surface morphology and topography-governed anti-frosting performance of the functional surfaces is not new - https://www.sciencedirect.com/science/article/pii/S1359431116338388?via%3Dihub ; https://www.sciencedirect.com/science/article/pii/S0017931016327673?via%3Dihub 

4) How did you determine the amount of accumulated water? This is not clear to the reviewer. 

5) It is claimed that self-propelled coalescence-jumping of subcooled condensates is the main anti-frosting mechanisms of the as-prepared surfaces, but there are no direct evidences shown in the text. It is suggested to add high-quality images (high magnification and resolution), where this can be clearly demonstrated. Also, Boreyko et.al. have explained the above mechanism in detail, so it seems there is nothing new here - https://aip.scitation.org/doi/10.1063/1.3483222 ; https://journals.aps.org/prl/abstract/10.1103/PhysRevLett.103.184501 

6) Apart of surface morphology, the surface chemistry plays a crucial role in the halting of interdroplet frost wave. For example, reducing the number of hydrophilic active sites on carbon materials leads to delayed frost formation. This approach should be at least mentioned and the relevant literature cited. 

7) Why the ACA and RCA are considered instead of the static contact angle? The previous are used to define the range of contact angles a droplet can fall in without changing the length of the three-phase contact line. Small difference between ACA and RCA means low contact angle hysteresis and high droplet mobility. You have an instrument from DataPhysics, Germany, so you must measure both, the static contact angle and the contact angle hysteresis. 

Author Response

Thank you for your comments concerning our manuscript. We have studied comments carefully and have made the correction. We hope it meets with approval. The main corrections in the paper and the responds to the editor’s comments are listed below point by point as follows:

Response to Reviewer 4 Comments

Comment 1:

Using laser beam for modifying the surface morphology is not new and in fact, it is quite a well-known technique (https://www.mdpi.com/1996-1944/13/24/5692 ; https://www.mdpi.com/2079-6412/11/1/58 - many more can be found in the scientific databases). From that point-of-view, this research is an incremental scientific advance.

Response 1:

Thanks to the reviewer. I definitely confirmed that this manuscript is an incremental scientific advance. The technology of laser surface modification is now widely studied, and its applications such as wettability, spectral reflectance, anticorrosion are extensively researched by many people. Thus, we know that this work is not a pioneering creation on fabrication an anti-frosting surface via ultrafast laser processing, but an incremental contribution which deliver a novel method to optimize the processing parameters is carried out. Nevertheless, the compatibility on different metallic materials is also validated. The work provides insight understanding of fabricating a proper superhydrophobic surface for anti-frosting, and tells practical parameter framework for this purpose. From the aspect of surface self-propelled droplet jumping, we give a novel way to study anti-frosting phenomenon. The jumping event and its record experimental principle were stablished as well.

In conclusion, we introduced a novel methodology for anti-frosting surface via laser-processed, and validated its compatibility for other metallic materials. Bridges were constructed to connect the parameters and practical performance. Experimental principle was also set out that immediately detect the surface by analyzing the jumping event. The experimental method, fabricating technology, optimizing guideline and the deep explanation for anti-frosting mechanism, all of these do make contribution to promote the laser modified functional surface technology and its applications. An incremental scientific advance is made based on this work.

Comment 2:

The authors claim that there is a lack of literature sources dealing with optimization of the laser parameters for tuning the icephobic properties of the surface, which is also not true (https://www.mdpi.com/2079-4991/11/1/135 ; https://www.scientific.net/KEM.926.1643 - and many more in the scientific databases). Based on 1) and 2), the reviewer strongly recommends the authors to deeply examine the current state-of-the-art, cite the mentioned and other relevant literature, and then clearly outline what the main novelty of their research is.

Response 2:

Thanks to the reviewer. Firstly, we feel sorry for rising a confusion of the express on literature sources with respect to the topic. The according content is revised. Obviously, many researches dealing with the optimization of laser parameters for tuning the surface performance, like superhydrophobicity and anti-frosting. However, major of these literatures were focus on a few pure metals, such as aluminum, copper, zinc, rare work on alloy, especially in anti-frosting surfaces. Alloy is widely used in many fields rather than the pure metal, thus for the prospect and practicability of this technology, the research on alloys is desired to be executed. The alloy of copper-zinc is rarely reported before, and it is widely used in the fields of electronics, fuel cell system and aerospace. In these bases, we performed our research on the copper-zinc alloy, for which the method would be examined on alloy surface. This work would break the barriers on the development of metallic anti-frosting materials and further applications.

Secondly, we found that the current state-of-the-art is effect on other metallic materials. We extracted a relationship between droplet jumping features and laser processing parameters from series of experiments, and further connected to the anti-frosting performance. On our best known, the research route which firstly construct the relationship between surface droplet jumping capability and laser processing parameters, then extract the outline for the optimization is rarely reported by other researchers. Others who worked on this topic always delivered limited data and results to comprehensively detect the surface anti-frosting performance. The research route, experimental principle, framework for the optimization, all these advances tell the novelty and scientificity of this study. In addition, with the material compatibility, our guideline characterizes universal and practical properties for many metallic functionalization in anti-frosting.

In conclusion, the revision version has reclaimed the main novelty by revising the according content and citing relevant literatures.

Comment 3:

The surface morphology and topography-governed anti-frosting performance of the functional surfaces is not new - https://www.sciencedirect.com/science/article/pii/S1359431116338388?via%3Dihub ; https://www.sciencedirect.com/science/article/pii/S0017931016327673?via%3Dihub.

Response 3:

Thanks to the reviewer. As the comment 1 claimed that this work is an incremental scientific advance. We all follow with this point. The field of surface morphology and topography-governed anti-frosting performance of the functional surfaces is extensively researched for over 13 years. It is indeed not a new field. The literatures ‘Boreyko, J. B. and C.-H. Chen (2009). "Self-Propelled Dropwise Condensate on Superhydrophobic Surfaces." Physical Review Letters 103(18): 184501.’ and ‘Enright, R., et al. (2014). "How Coalescing Droplets Jump." ACS Nano 8(10): 10352-10362.’ and ‘Miljkovic, N., et al. (2013). "Jumping-Droplet-Enhanced Condensation on Scalable Superhydrophobic Nanostructured Surfaces." Nano Letters 13(1): 179-187.’ and ‘Zhang, Q., et al. (2013). "Anti-icing surfaces based on enhanced self-propelled jumping of condensed water microdroplets." Chemical Communications 49(40): 4516-4518.’, all these reported that the concept of functional surface design, mechanism of self-propelled droplet on textured surface, the according theory and principle to fabricate such surface, were comprehensively researched. However, the technology is not practical and mature currently. Many issues are going to be deal with, such as its durability, precious controlling, material compatibly, deep mechanism of effects between surface texturing and functionalization. The technology confronts these barriers, therefore limits the development. People who involve the field are going to make contribution to break these concerns for its practical applications. So, as the replies 1 and 2 claim that, this work tells a small step to progressively construct the bridges between laser processing parameters, wettability, droplet jumping capability and anti-frosting performance. Moreover, the several other metallic materials were detected based on the processing outline we extracted. We believe the results give valuable advice to fabricate such functional surfaces with the universal method. We believe that it makes sense to propel the practical application in this field.

In conclusion, the provided articles are added to the manuscript and made some revisions accordingly.

Comment 4:

How did you determine the amount of accumulated water? This is not clear to the reviewer.

Response 4:

Thanks to the reviewer. The amount of accumulated water was determined by manually calculating the droplet diameter in the software of Nanomeasure and Image J, and then calculating the weight by multiplying the water density. As shown in Figure 2, the photograph of a droplet is spherical cap. We assume that the droplets are spherical since the contact angle is larger than 150°. Notably, the calculated droplet volume under such assumption has less error than 5% that calculated with the geometry of spherical cap. The schematic illustration of this experimental setup is shown below.

This method was also used in the previous article (references, number 5). Zhao, G., et al. (2020). "Rationally designed surface microstructural features for enhanced droplet jumping and anti-frosting performance." Soft Matter 16(18): 4462-4476.

In conclusion, we provide new reference and add more explanations for this comment.

Comment 5:

It is claimed that self-propelled coalescence-jumping of subcooled condensates is the main anti-frosting mechanisms of the as-prepared surfaces, but there are no direct evidences shown in the text. It is suggested to add high-quality images (high magnification and resolution), where this can be clearly demonstrated. Also, Boreyko et.al. have explained the above mechanism in detail, so it seems there is nothing new here - https://aip.scitation.org/doi/10.1063/1.3483222; https://journals.aps.org/prl/abstract/10.1103/PhysRevLett.103.184501.

Response 5:

Thanks to the reviewer. The main mechanism of anti-frosting on a superhydrophobic surface is validated and recognized by many researchers. It is a generally acknowledged principle in designing anti-frosting nanostructure on superhydrophobic surfaces. So, as the previous relies say claimed that this work is an incremental scientific advance on this topic. This work tells a small step to progressively construct the bridges between laser processing parameters, wettability, droplet jumping capability and anti-frosting performance. Moreover, the high-quality images for the mechanism of main anti-frosting based on self-propelled coalescence-jumping were already provided. Let`s return to the Figure 2 in the manuscript, the picture gives the illustration about the wetting properties of sessile macroscale and condensed microscale droplets on the different surfaces. What is more, the Figure 5 in previous literature of our group ‘Zhao, G., et al. (2020). "Competing Effects between Condensation and Self-Removal of Water Droplets Determine Antifrosting Performance of Superhydrophobic Surfaces." ACS Applied Materials & Interfaces 12(6): 7805-7814.’ illustrated distinctly that the amount of water removed from and accumulated on the surface is govern by different surface morphology we designed. To explain the mechanism in detail, we provided a series of theoretical analysis on the effect of wetting property on coalesced droplet jumping capability (S3, in supplementary materials). These analyses were based on the aspect of adhesion energy in the contact system. In this section of supplementary materials, Figure S6 tells the mechanism of droplet jumping capability over three different wetting state. Moreover, Figure S7 tells the surface with Cassie-Baxter wetting state enables the better droplet jumping capability than partial wetting, and the smaller droplet can jump easier. All the aforementioned contents and figures can back up the main anti-frosting mechanisms in our work.

Comment 6:

Apart of surface morphology, the surface chemistry plays a crucial role in the halting of interdroplet frost wave. For example, reducing the number of hydrophilic active sites on carbon materials leads to delayed frost formation. This approach should be at least mentioned and the relevant literature cited.

Response 6:

Thanks to the reviewer. The surface chemistry is crucial in water special wetting applications. It is well known that the surface with less hydrophilic sites (more hydrophobic sites) enables a stronger capability of water repellence. Thus, combination of the surface texture and a low surface energy process was employed in this work. The hydrophilic sites on the surface were already reduced by chemical modification (FAS-17), hence a long-chain molecular layer with hydrophobic site was fabricated to lead to superhydrophobicity. This layer introduces the potential of droplet jumping capability, and consequently the functionalized surface can be designed and optimized to a better performance of anti-frosting.

To enhance our demonstration on this issue, a relevant literature (Esmeryan, K.D.; Fedchenko, Y.I.; Yankov, G.P.; Temelkov, K.A. Laser Irradiation of Super-Nonwettable Carbon Soot Coatings–Physicochemical Implications. Coatings 2021, 11, 58.) which tells the relationship between hydrophilic active sites and frost formation procedure is cited.

Comment 7:

Why the ACA and RCA are considered instead of the static contact angle? The previous are used to define the range of contact angles a droplet can fall in without changing the length of the three-phase contact line. Small difference between ACA and RCA means low contact angle hysteresis and high droplet mobility. You have an instrument from DataPhysics, Germany, so you must measure both, the static contact angle and the contact angle hysteresis.

Response 7:

Thanks to the reviewer. Apart from the ACA and RCA, the static contact angles were also detected in our experiments. We found that the static contact angles were close to the ACA during the wettability experiment. Although the static contact angle is a basic aspect in contact angle measurements, it tells less information than the ACA and RCA. As the words in comment 7 said that the ACA and RCA are used to define the range of contact angles that a droplet can fall without changing the three-phase contact line and the difference between the previous show surface capability of the droplet mobility, we think the ACA and RCA are more direct and scientific and proper results to determine the droplet jumping capability. What is more, we had interpreted that the static contact angles of all these substrates are over 150°. In detail, we found all the surface exhibit superhydrophobicity after chemical modification with almost the same static contact angle over different samples. So, the static contact angle can hardly affect in determine the droplet jumping capability in this work. Since the phenomenon of different wetting states, Cassie-Baxter state, Wenzel state and mixture state were found in different group of surfaces, ACA, RCA and sliding properties were more effective data for supporting the topic. This manuscript is studied based on observing the self-propelled coalescence induced jumping of micro-droplets on the subcooled surfaces, thus repellence capability against a bigger droplet is worthless. It is a common sense that gravity affects weaker in contact angle measurements of a micro-droplet than the surface tension force. On these bases, we can barely learn any more information from static contact angle, but the ACA and RCA can.

We tried our best to improve the manuscript and made some changes in the manuscript. These changes will not influence the content and framework of the paper.

Once again, thank you very much for your comments.

Best regards,

Jianguo He

Round 2

Reviewer 4 Report

The following comments apply to the second version:

1) Taking into account that the manuscript has been refereed by 4 reviewers, the incorporated changes in the text are very minor and obviously, the article has been very lightly revised. 

2) Providing a paper with track changes on is not a good approach and causes confusion. Normally, the corrected/revised text must be highlighted in yellow. 

3) I am not sure whether a paper showing incremental scientific advance (as confirmed by the authors too) should be published in a journal category Q1. Stating that the novelty is related to the use of alloy instead of pure metal is not serious. 

4) Regarding the self-propelled coalescence and droplet jumping - if your paper does not contain sufficient amount of new analysis and concepts concerning this physical mechanism, it will be more or less unusable. 

5) There is miscited scientific literature. For example, ref. 20 has nothing to do with "metallic SHSs via chemical oxidation or electrochemical deposition".

Author Response

Thank you for your comments concerning our manuscript. The responds to the comments are listed below point by point as follows:

Response to Reviewer 4 Comments

Comment 1:

Taking into account that the manuscript has been refereed by 4 reviewers, the incorporated changes in the text are very minor and obviously, the article has been very lightly revised.

Response 1:

All of the comments have been addressed and revised to their proper form. We provided detailed responses. We do not think we need more revision with respect to the novelty and physical mechanism of current form. Combining with the supplementary materials of data and videos, we can state that this work is complete and systematical for the anti-frosting applications on alloy materials. Any other progressive step to propel the research will start new work, such as the durability of the structured layer, fabrication efficiency of the large-area functional layer, and non-chemical-modified superhydrophobic surfaces. Although these will provide the extensive prospect for the application, they are not down to the scope of this work. We have come up with a methodology of quick fabrication, optimization, and experimental inspection for this technology thoroughly. Therefore, we have made enough changes in the text to improve the manuscript.

Comment 2:

Providing a paper with track changes on is not a good approach and causes confusion. Normally, the corrected/revised text must be highlighted in yellow.

Response 2:

Thanks to the reviewer. The corrected/revised texts are highlighted in yellow.

Comment 3:

I am not sure whether a paper showing incremental scientific advance (as confirmed by the authors too) should be published in a journal category Q1. Stating that the novelty is related to the use of alloy instead of pure metal is not serious.

Response 3:

In the field of spontaneous condensed microdroplet jumping, other than a few pioneer works such as “Self-propelled dropwise condensate on superhydrophobic surfaces”, which first observed droplet jumping phenomenon and proposed the mechanism; “Tuning Superhydrophobic Nanostructures To Enhance Jumping-Droplet Condensation”, “Effect of droplet morphology on growth dynamics and heat transfer during condensation on superhydrophobic nanostructured surfaces”, and “Droplet jumping: effects of droplet size, surface structure, pinning, and liquid properties”, which analyzed droplet jumping behavior in great detail; “Delayed frost growth on jumping-drop superhydrophobic surfaces”, and “Anti-icing surfaces based on enhanced self-propelled jumping of condensed water microdroplets”, which introduced droplet jumping to anti-icing applications; “Jumping-droplet-enhanced condensation on scalable superhydrophobic nanostructured surfaces” and “Condensation heat transfer on superhydrophobic surfaces”, which introduced droplet jumping to heat transfer applications; “Antifogging abilities of model nanotextures”, which introduced droplet jumping to antifogging applications; “Nanograssed micropyramidal architectures for continuous dropwise condensation” and “Hierarchical Porous Surface for Efficiently Controlling Microdroplets' Self‐Removal”, which introduced hierarchical micro-nanostructure onto droplet jumping surface to enhance performance; “Coalescence-induced jumping of droplets on superomniphobic surfaces with macrotexture”, “Self-Propelled Droplet Removal from Hydrophobic Fiber-Based Coalescers”, “Particulate–Droplet Coalescence and Self-Transport on Superhydrophobic Surfaces”, and “Laplace pressure driven single-droplet jumping on structured surfaces”, which really expand the field of droplet jumping and opens more opportunity for future applications; all other 1000 or so related papers (give and take) on spontaneous condensed microdroplet jumping are more or less “incremental” scientific advance. Hence, we really don’t think that contributing incrementally in the field of condensed droplet jumping is a problem.

In terms of our novelty, we can state clearly here:

  1. a) We provided a quick and facile fabrication strategy for spontaneous condensed microdroplet jumping anti-frosting surface with excellent performance on possibly any metallic surface using ultrafast laser technology, which was not achievable using chemical or electrochemical techniques.
  2. b) We provided a quick but effective optimization strategy for anti-frosting surfaces by establishing a concrete relationship between laser irradiation parameters, condensed droplet jumping capability, and anti-frosting performance.

Metallic anti-frosting surfaces have a wide range of applications, for instance, titanium and aluminum alloys are commonly used on the wings of aircraft, which are in dire need for anti-frosting/icing surface treatment, and it would be much more desirable to find a simple fabrication method. Anyone with expertise in the field of anti-frosting should be able to appreciate our work. It is not “novel” as compared to the works we have pointed out above, but it is of great importance if droplet jumping surfaces are to be utilized in future engineering applications.

Therefore, we insist that our manuscript has enough novelty and is publishable, which is verified by the positive comments from all other 3 reviewers. Also, judging from the review comments (e.g. static contact angle vs advancing/receding contact angle), we think that Reviewer #4 lacks expertise in the field of spontaneous condensed droplet jumping and anti-frosting to provide a fair evaluation on the novelty of our work. Moreover, judging from the difference in Review #4’s marking scheme, we think he/she took our previous reply personally, and cannot provide an objective review anymore. If possible, we tentatively request the editor to change Review #4 to someone with better expertise.

Comment 4:

Regarding the self-propelled coalescence and droplet jumping - if your paper does not contain sufficient amount of new analysis and concepts concerning this physical mechanism, it will be more or less unusable.

Response 4:

We have provided enough statements on the contribution of this work. Please see response #3.

Comment 5:

There is miscited scientific literature. For example, ref. 20 has nothing to do with "metallic SHSs via chemical oxidation or electrochemical deposition".

Response 5:

Thanks to the reviewer. The description of the miscited literature is revised..

Once again, thank you very much for your comments.

Best regards,

Jianguo He